# Examining retweeting behavior on social networking sites from the perspective of self-presentation

**Juan Shi** [ORCID][1]*, **Kin Keung Lai**[1], **Gang Chen**[2]

1 International Business School, Shaanxi Normal University, Xi'an City, Shaan'xi Province, China,
2 Shanghai Huace Navigation Technology Ltd., Shanghai City, China

* shijuan4258@qq.com

**Data Availability Statement:** All relevant data are within the manuscript and its Supporting Information files.

**Funding:** This work was supported by the Natural Science Basic Research Program of Shaanxi (Program No. 2020JQ- 427) and the Fundamental

## Abstract

On social networking sites, people can express themselves in a variety of ways such as creating personalized profiles, commenting on some topics, sharing their experiences and thoughts. Among these technology-enabled features, retweeting other-sourced tweet is a powerful way for users to present themselves. We examine users' retweeting behavior from the perspective of online identity and self-presentation. The empirical results based on a panel dataset crawled from Twitter reveal that, people are prone to retweet topics they are interested in and familiar with, in order to convey a consistent and clear online identity. In addition, we also examine which user groups exhibit a stronger propensity for a clear online identity, considering the practical value of these users to both social media platforms and marketers. By integrating self-presentation theory with social influence theory and social cognitive theory, we propose and confirm that users with higher value in online self-presentation efficacy and users who are more involved with the social media platform have a stronger than average propensity to maintain a consistent online identity, and thus are more likely to retweet familiar topics. These users are characterized by (1) owning a larger number of followers, (2) authoring longer and more original tweets than average, (3) being active in retweeting other-sourced posts. This study contributes to our understanding of SNS users' retweeting behavior and adds to the emerging line of research on online identity. It also provides insights on how microblogging service providers and enterprises can promote people's retweeting behavior.

## Introduction

Individuals are fundamentally motivated to present themselves in everyday life [1]. In offline environments, people present their tastes and identities via attitudes, preferences and visible products such as cars, clothes, music genre [2] and so on. With the advent of web 2.0, individuals are able to manage their online images in new and exciting ways [3]. They purchase digital items [4], broadcast their experiences, share their thoughts and opinions in the form of words, photos or videos, or interact with others by "comment", "like". These technology-enabled

Research Funds for the Central Universities (Program No. 21SZYB24). The funders had no role in study design, data collection and analysis, decision to publish, or preparation of the manuscript.

**Competing interests:** The authors have declared that no competing interests exist.

features allow users to represent and express their identities to others [5], thereby promoting mutual understanding in computer-mediated contexts [6]. Such understanding facilitates smooth and conflict-free online interactions and thus can improve relationship formation and information exchange between online users [7, 8].

Among those technology-enabled features, retweeting other-sourced tweet is a powerful way for SNS users to express themselves [9, 10]. Just by clicking on the "Retweet" button on Twitter, all this user's followers will see the shared content immediately. Theoretically, a user has the opportunity to cultivate whatever identity they desire through the content they share. However, [11] confirm that, when making forwarding decisions, users tend to focus on a limited span of topics to construct a consistent and clear online identity, which distinguishes themselves from others, helps to find similar ones and establish relationships [12]. This phenomenon is more conspicuous among "expert" users, who employ Twitter as a personal branding tool and strive to signal a consistent online image to a greater extent [13]. Their posts (whether self-produced tweets or retweets) fit like-minded users' expectations and can attract more and more followers on SNS. As a result, the reputation and influence of these "expert" users are further promoted.

We argue that users with a more consistent and clear online identity are valuable for both social media platforms and marketers who wish to employ influencer marketing and engage other SNS users in voluntary electronic word-of-mouth (eWOM) communication. Three reasons underlie this claim. First, these users have a higher level of loyalty and participation to the social media platform [7, 14], as they have established a desired online image on the platform [15, 16], bringing them recognition and an amplified sense of self-worth [17, 18]. Second, these users have the potential to become key opinion leaders (KOLs) and attract plenty of like-minded followers [19]. Virtual communities may be formed revolving around these KOLs, within which information exchange and relationship building will be greatly facilitated [8]. Third, users with a consistent and clear online persona can be considered as endorsers in influencer marketing [20]. Marketers can choose users whose online identities match with the traits of their products. Researchers have proven that high influencer-product congruence induces high consumer-product congruence, which produces higher purchase and recommendation intentions [21].

Despite the practical value of these users, there is a lack of understanding about which user groups exhibit a stronger proclivity for a consistent and clear online identity, let alone how to identify and retain these users. Previous research has mainly investigated the effect of online identity on individuals' online behaviors such as purchasing digital items [19], customer citizenship behavior [22, 23] and social commerce engagement [24]. Of particular interest in the literature is the study by [11], who verify that professional bloggers are more likely to convey a consistent and focused online identity on Twitter. However, in addition to these professional bloggers, which user groups have a stronger tendency for a focused online identity is still unclear. Put differently, the characteristics of these users have not been studied in a broader sense. This gap and the practical value of these users motivate our research that aims to examine SNS users' retweeting behavior from the perspective of online identity and self-presentation to provide specific guidelines for identifying users who have a more focused online identity.

Researchers have confirmed that retweeting familiar topics is beneficial for establishing a consistent and focused online identity [11]. Therefore, we examine the relationship between topic familiarity and individuals' retweeting behavior and investigate what factors moderate this relationship. By doing this, we are able to identify users who are more likely to maintain a consistent online image through retweeting familiar topics. According to self-presentation theory, individuals tailor their public images to different social settings and to the extent of their

control on the presentation. Thus, both social influence (e.g., social norms [25], virtual community involvement [4]) and personal control (e.g., online presentation self-efficacy [4]) could affect people's desire for a consistent online image. For social influence, we use "retweet interval" and "number of original tweets" to denote individuals' level of involvement to the social media platform [26]. Gender differences are also examined as self-presentation norms specify that men and women should convey different impressions. For personal control, we use "number of followers" and "average tweet length" to indicate individuals' efficacy of self-presentation [1, 27].

The rest of the paper is arranged as follows. In the following section, we introduce our main theoretical foundation and related studies. In Section 3, we develop our research hypotheses and propose a concept model. In Section 4, data collection, measurement of variables and the regression model are presented. In Section 5, we elaborate on regression results and check the robustness of our results. Finally, we discuss our research findings, theoretical and practical implications and provide directions for future research in Section 6.

## Literature review

We first elaborate on the concept of online identity and self-presentation theory and review related research on these theories. And then we identify the social influence and personal control constructs that are used as moderators in our conceptual model.

### Online identity

Online identity refers to the configuration of the defining characteristics of a person in online space [28], which distinguishes this person from others and makes the person feel distinctive and unique. Social networking sites (SNS) allow users to project a desired online identity using various digital items such as avatars, nicknames, virtual badges and so on [7]. In the Stack Overflow question and answer community, earned badges help contributors signal their unique identity and efficiency, because earned badges reflect owners' skills and interests, and therefore, convey their identity to other users in the community [29].

Identities are used to help organize and direct people's lives and therefore inform a variety of behavior decisions. Specifically, individuals have a need for coherence and for this reason they seek out objects (e.g., products, political ideology) that provide feedback consistent with their self-concept [30]. For example, consumers are attracted to, and more likely to purchase, identity-consistent products to affirm important self-concepts [31]. A research on bloggers shows that identity congruence between identity expectation and perceived identity performance in blogging determines their willingness to continue using the blogging service [5]. In social media, although the retweet function provides twitterers with an opportunity to share any content to construct a knowledgeable and diverse online image, it has been validated that users prefer to retweet topics which they are able to self-produce [11], they are adept at [32], or they are interested in [10, 33], as sharing familiar topics helps to construct a clear and focused online identity.

Although prior studies have advanced our understanding of SNS users' online identity construction behavior, which user groups exhibit a stronger propensity for a consistent online identity remains unknown. Considering the practical value of these users both to social media platforms and to enterprises, we investigate SNS users' retweeting behavior from the perspective of online identity and self-presentation and identify these users by analyzing what moderators affect the influence of topic familiarity toward individuals' retweeting behavior on SNS.

## Self-presentation theory

Self-presentation or impression management refers to the process of influencing how one is perceived by others [1, 25]. In most situations, people engage in impression management because they believe, rightly or wrongly, how others respond to them is largely determined by the impressions they make on others [34]. When a candidate chooses the clothes he will wear on a job interview, students clean up their dormitory when the dorm supervisor is coming to check, individual sellers on Airbnb adopt social-oriented presentation strategy to boost perceived trustworthiness and booking intentions [6], their behavior is explicitly motivated by self-presentational concerns.

However, people sometimes impression manage even when there are no reasons to influence others' perceptions or behaviors. Self-presentation theory posits that in such situations, people are constructing and communicating a specific and desired identity to others, also called "identity signaling" in consumer behavior literature [35]. For example, when users exert themselves to present an attractive, professional or humorous personal homepage although most visitors are probably strangers and not familiar with them at all [36], consumers actively interact with brand-related elements in a brand community to convey their identities [37], individuals choose products with symbolic or expressive value (e.g., car brand, hairstyle, music genre) to signal their tastes [38], they are highlighting their unique characteristics which define their identities and distinguish themselves from others.

According to self-consistency theory [39], people have a fundamental motivation to maintain consistency in self-representation. A clear and consistent online identity on social networking sites reflects one's interests and uniqueness [40], helps others understand the kind of people you are and assess whether there is a need to connect with you [41]. Through sharing familiar and relevant topics [10], most twitterers enrich the depth of their online identity to make it more cohesive and consistent, and this propensity is more conspicuous in professional bloggers [11]. However, in addition to professional bloggers, which user groups have a stronger tendency for a focused online identity is still unclear and requires further investigation on this issue.

## Social influence and personal control

Individuals' self-presentation behavior is a function of both the immediate social context and the person's degree of control over his or her impression management [25]. Therefore, both social influence [42, 43] and personal control [4, 44] have been suggested to affect one's self-presentation behavior, or the construction of a consistent online identity via retweeting in the context of our study.

[45] has proposed three processes of social influence: compliance, internalization, and identification. Compliance represents a process in which people adhere to the formal and informal norms of the system in order to gain social approval or to avoid punishment. Formal rules for self-presentation are not likely to exist in the context of SNS. Instead, social norms which suggest that men and women should present different public impressions are likely to operate. Men are encouraged to present a powerful, ambitious, and competent impression to others, whereas women's self-presentations emphasize their interpersonal and communal attributes [46]. As a result, men's self-presentation on SNS accentuates status, achievement and power, whereas women's self-presentation on SNS accentuates familial relations and emotional expression [47, 48]. Therefore, we speculate that men and women will adopt different retweeting strategies to construct different desired images on Twitter.

In addition, identification occurs when an individual accepts influence from others to maintain a satisfying self-defining relationship to others [49]. Identification with a particular

social context (e.g., an online community) has been shown to have a positive impact on one's desire for self-presentation [4, 23]. When individuals perceive themselves as members of a virtual community, they will have a positive attitude toward the community [49], exhibit prosocial behaviors in the group [23], and have a higher level of participation and involvement with the group [50]. Thus, individuals' degree of identification with a community can be inferred from their level of involvement and participation in the community [4, 44]. Consistent with [26], we use "retweet interval" and "number of original tweets" to infer one's level of involvement with Twitter. They verify that the more actively a user exhibits on Twitter, the more frequently (s)he uses the function of Twitter such as tweeting, retweeting, or sending an @Reply, and finally, the more gratification the user derives.

Personal control over one's self-presentation, also called impression efficacy in [25] or self-presentation efficacy in [4], is an optimistic belief about one's own ability in building desired impressions on others [51]. To the extent that online self-presentation can be a challenging task, individuals' sense of impression efficacy should influence their desire and performance in online impression management. People with high self-presentation efficacy have a stronger desire to present themselves [4, 44], display a larger number of friends [11], and have formed their own personal style [52]. Moreover, longer contents are indicative of careful thinking and contain more information [53], serve better in building one's online image [1], and thus are more likely produced by users who care more about their online images and are confident in achieving this goal. Therefore, for personal control, we use "number of followers" and "average tweet length" to indicate individuals' online self-presentation efficacy.

## Theoretical model and hypotheses

### Topic familiarity and individuals' sharing behavior

Fig 1 illustrates our conceptual model. The independent variable is topic familiarity that is the degree to which a tweet is similar or relevant to an individual's interested topics. The dependent variable is an individual's response to a post, with 1 denoting this user shared this post and 0 otherwise.

From the perspective of self-presentation, sharing a post on SNS is not only spreading it to more people but also an effective way of expressing oneself. By sharing familiar and relevant topics, SNS users can enrich the depth of their online identity and make it more consistent and cohesive. What's more, motivated by their online identities, users often allocate greater attention to identity-consistent messages and shift attention away from identity-inconsistent contents [54]. Therefore, messages with familiar topics are more likely to attract one's attention and have a larger chance of being shared. In support of this, existing research has shown that on Twitter, individuals tend to retweet topics they are adept at [32], they are able to self-produce [11] or they are interested in [10, 33]. And this phenomenon is more prominent among expert users who strive to exhibit a consistent online identity to a greater extent [11]. Hence, on social networking sites, a higher degree of topic familiarity should lead to a larger likelihood of being shared for the message.

**Hypothesis 1** *The more familiar a tweet is perceived, the more likely an individual will retweet it to all his or her followers on SNS to build a consistent online identity.*

### Moderating effects

**Online self-presentation efficacy.** According to social cognitive theory [51], self-efficacy is a feeling of competence and effectiveness with regard to a certain task. Its specific form in our study is online self-presentation efficacy which refers to individuals' confidence in their abilities to present desired images on Twitter. [27] have validated that online self-presentation

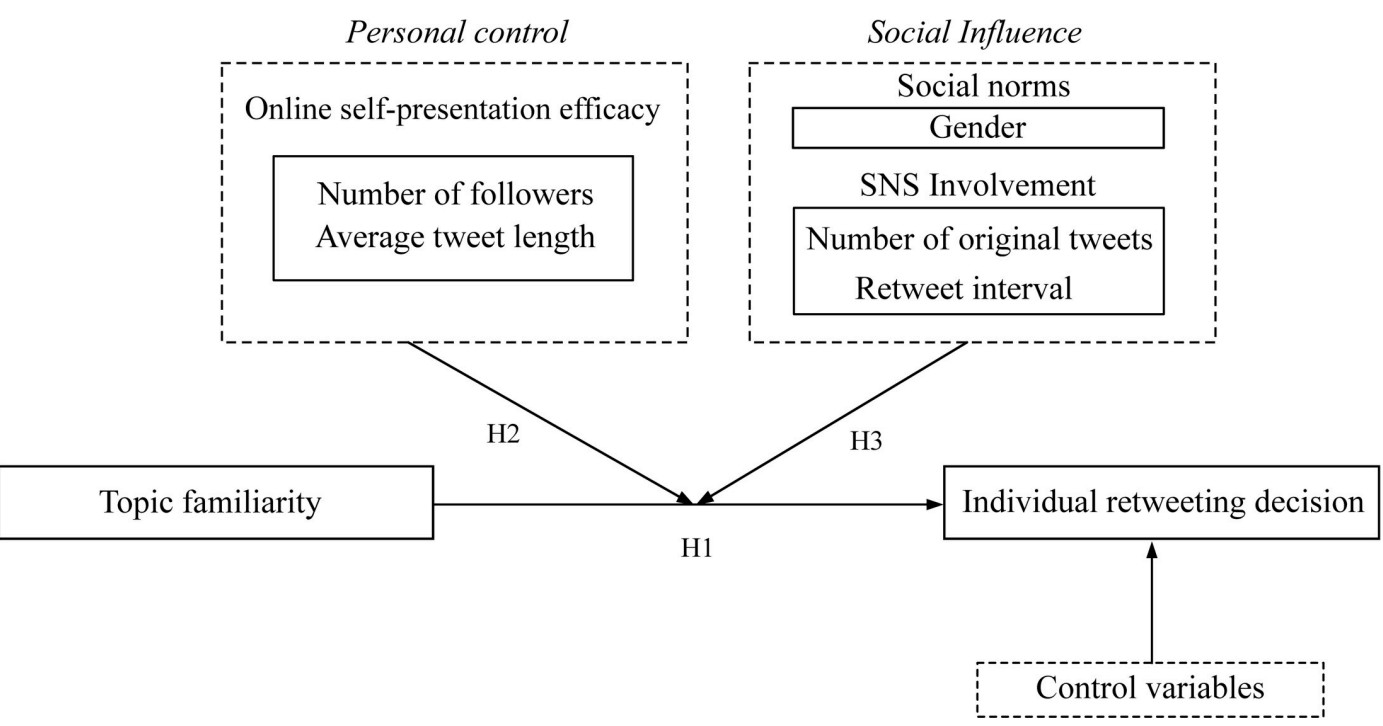

**Fig 1. Conceptual model.**

efficacy is significantly related with the number of friends and the degree of profile detail. Therefore, we use "number of followers" and "average tweet length" to infer an individual's online self-presentation efficacy.

*Number of followers.* From the perspective of impression management, twitterers with a larger quantity of followers have spent more time and effort on constructing and maintaining their online identities than casual users (casual users are representative of the Twittersphere population). As the number of followers increases, these twitterers become more skillful in interacting with their followers, more adept at using Twitter to communicate and highlight their distinctive value, and finally, they manage to establish a clear online identity on social networking sites. Therefore, the number of followers is an informative social signal implying the user's influence as well as their ability and experience in conveying desired images [55].

Put differently, popular twitterers score higher in online self-presentation efficacy and tend to demonstrate superior performance in this task [27]. When making retweeting decisions, they are more inclined to convey a clear and cohesive online identity to highlight their uniqueness. [11] have confirmed that professional bloggers, who often have a larger number of followers, strive to signal a focused online persona through limiting the scope of topics to familiar and relevant ones in the shared content. Given these arguments, we hypothesize:

**Hypothesis 2a:** *The number of followers positively moderates the influence of topic familiarity on individuals' sharing behavior. Specifically, an increase in topic familiarity leads to a larger likelihood of sharing the tweet for users who have more followers than for users who have less followers.*

*Average tweet length.* Average tweet length is the average number of words in original tweets contributed by a focal user until the time he makes the retweeting decision. Users who are used to producing longer original contents are willing to expend more time and cognitive effort to make the contents informative and eloquent, thereby creating an influential and

impressive online image [1]. In contrast, users who are prone to produce shorter contents have no intention or ability to provide lots of information, and fail to construct as impressive and clear online images as those "diligent" users. Thus, we infer that "diligent" users are confident in elaborately expressing themselves online, and thus have a stronger feeling of self-efficacy in online self-presentation [51]. In support of this, [27] have validated that on a popular social networking site in German, the profiles of individuals with higher self-presentation efficacy are more elaborate in terms of completed fields and number of words.

Considering the fact that a clear and consistent online image facilitates mutual understanding and relationship building among users [5], we conjecture that "diligent" users, who know better how to use Twitter as a self-presentation tool, are more likely to share familiar topics to establish a clear online identity, thereby distinguishing themselves from others.

**Hypothesis 2b:** *Average tweet length positively moderates the influence of topic familiarity on individuals' sharing behavior. Specifically, an increase in topic familiarity leads to a larger likelihood of sharing the tweet for individuals who author longer tweets than for users who author shorter tweets.*

**Social influence.** *Social norms*: *Gender*. Prior research reveals that online images constructed by men and women are quite different on social networking sites [48]. Specifically, men's self-presentation accentuates status, achievement and power which makes them more attractive, whereas women's self-presentation accentuates familial relations and emotional expression. These gender differences can be explained from the perspective of social norms, which specify that men and women should present different public impressions. Men should present themselves as competent and powerful, whereas women should underlie their interpersonal and communal attributes [25].

Following the similar logic, we conjecture that men are more motivated than women to broaden the span of topics when making retweeting decisions, thereby creating a competent, information-rich and diverse online identity on SNS. By doing this, they can appeal to a wider range of friends, have more opportunities to build relationships with new friends, and finally promote their popularity and reputation. Whereas for women, a more focused and authentic online identity helps to express the true self and maintain long-term relationships with their existing social network [56]. Given these arguments, we hypothesize that compared with men, women are more likely to limit the scope of shared contents to present a more focused online identity.

**Hypothesis 3a:** *The influence of topic familiarity on individuals' sharing behavior is stronger for female twitterers than for male twitterers. Specifically, an increase in topic familiarity leads to a larger likelihood of sharing the tweet for female twitterers than for male twitterers.*

*SNS Involvement*: *Number of original tweets*, *Retweet interval*. SNS involvement refers to the degree to which users indulge in social media and indicates how active users are [57]. Consistent with [26], we use "number of original tweets", which were authored by the user in the past six months, and "retweet interval", measured by the number of days that have elapsed since the user's last retweeting, to infer one's level of SNS involvement. Accordingly, active users have a larger number of original tweets and shorter retweet intervals due to their frequent activities on SNS.

By publishing original tweets, replying to others, and using other Twitter functions such as retweets, active users gratify their need to express their opinions, disclose their personal information, gain social support, and keep in touch with their online relationships [26]. These users utilize social media heavily, have developed a sense of belonging and communion with others [20], and thus have a higher level of involvement and identification with the platform. Therefore, these highly involved users are more motivated to construct a clear and impressive online image, which highlights one's competence, attitude, and preferences and thus can help others

interact smoothly with them [1]. In contrast, less involved users have a weak sense of belonging to the platform, and thus have no interest or intention to establish their online images.

Thus, we hypothesize highly involved users have a higher propensity to present a consistent and clear online identity by retweeting familiar or related topics. Since we use "number of original tweets" and "retweet interval" to indicate an individual's level of SNS involvement, we have the following two hypotheses:

**Hypothesis 3b:** *Number of original tweets positively moderates the influence of topic familiarity on individuals' sharing behavior. Specifically, an increase in topic familiarity leads to a larger likelihood of sharing the tweet for users who author more original tweets than for users who author fewer original tweets.*

**Hypothesis 3c:** *Retweet interval negatively moderates the influence of topic familiarity on individuals' sharing behavior. Specifically, an increase in topic familiarity leads to a larger likelihood of sharing the tweet for individuals who have shorter retweet intervals than for individuals who have longer retweet intervals.*

## Data and methodology

### Data collection

Twitter has received lots of attention from both academicians and practitioners [10, 11, 58]. Twitter users can publish updates in the form of words, pictures, music, videos, interact with others by "comment", "like" or "retweet" other-sourced tweets. Different with Facebook or LinkedIn, which requires reciprocity, Twitter users can follow any other accounts just by clicking "follow" as long as it is not a protected account. Via Twitter's streaming API, we obtained our core users according to the following steps.

1. During December 10, 2022 to December 13, 2022, we collected 50 trending topics on Twitter at a random moment each day, which resulted in 200 topics. We randomly select 50 topics from them and obtained 650,473 posts.

2. From the tweets collected in step (1), we draw uniformly at random a sample of 50,000 posts, which were produced by 42,019 Twitter users. Among these users, we screened out inactive users (users who retweeted less than 50 times or authored less than 50 self-tweets in the past 6 months) or non-English users, as our algorithm can only process English language. This process resulted in 23,342 potential core users.

3. From the set of potential valid users, we draw uniformly at random a sample of 1,500 users as the core users for our study. During December 20, 2022 to February 10, 2023, each core user's profile including the user's id on Twitter, whether this account is verified, the creation time of their account, number of followers and followees, and all the twitterer's tweets including self-produced tweets and retweets were crawled. During our collection window, a few users changed their privacy settings, meaning that they can be viewed only by approved followers. As a result, we obtained a final sample of 1,437 core users. Totally, we obtain 1,776,670 tweets published by these 1,437 core users.

We promise that the collection and analysis method complied with the terms and conditions for the source of the data, which is Twitter in the current work. We didn't divulge any information about the users, such as their IDs, topics of interest and so on in the current work. The processing and analysis of the dataset is limited to validating the hypotheses. All the data employed in this paper is obtained from Twitter via Twitter REST API. The Twitter API enables programmatic access to Twitter in efficient and easy ways.

## Measurement of variables

**Dependent variable $Y_{it}$.** $Y_{it}$ denotes the ith twitterer's reaction to the post encountered at time $t$. If this user retweeted the post, $Y_{it}$ was "1", otherwise "0".

**Predictors.** • *Topic familiarity*: $Topfami_{it}$

Due to the large size of our data set, it becomes unfeasible to ask each core user's opinion about his or her retweeting decision, as researchers do in questionnaire-based surveys. Thus, consistent with prior studies [10, 59], topic familiarity perceived by an individual about a specific tweet is measured by the degree to which the topics of the tweet are similar to this individual's interested topics. Specifically, using the Latent Semantic Analysis (LSA) topic modeling approach, we estimate topics contained in a tweet encountered at time t (denoted by $P_{tweet_t}$), and interested topics for the ith twitterer $s_i$ at time $t$ (denoted by $P_{s_{it}}$), and then calculate the cosine similarity between $P_{tweet_t}$ and $P_{s_{it}}$ to estimate topic familiarity, which is formulated in Eq (1). For detailed information of topics extraction process, please refer to [10].

**Definition 1** Topic familiarity perceived by the ith individual $s_i$ when he or she encounters a tweet at time t can be calculated as

$$Topic\_familiarity(tweet_t, s_{it}) = \frac{P_{tweet_t} \cdot P_{s_{it}}}{||P_{tweet_t}|| \cdot ||P_{s_{it}}||} \tag{1}$$

where $P_{tweet_t}$ represents the latent topics contained in the tweet encountered at time $t$, and $P_{s_{it}}$ denotes interested topics for $s_i$ at time t.

**Moderating variables.** • *Online self-presentation efficacy*: $Followers_i$, $AveTwtLen_{it}$

Compared with the amount of posts authored or shared by a twitterer, the number of one's followers usually changes slowly. Thus, $Followers_i$ is often treated as a time-invariant variable in prior research [60]. The descriptive statistics of our data show that there are great differences among the predictors in terms of their order of magnitude. For instance, $Topfami_{it}$ varies within the range [0,1], however, the maximum of $Followers_i$ is about six millions. As a result, the coefficients of predictors with a large magnitude are very close to 0 in the regression result. Therefore, we divide $Followers_i$ by 1.0e5 to make different predictors on the same scale.

Average length of tweet ($AveTwtLen_{it}$) refers to the "average number of words in each original tweet contributed by the $i$ th individual during the past six months before time $t$" [61].

• *Social norms*: $Gender_i$

We follow two steps to infer $Gender_i$,the gender of ith individual. First, we infer $Gender_i$ based on this twitterer's profile photo. Then, we double-check our inference using Gender API. It is the biggest platform on the internet to determine gender by a first name, a full name or an email address. When $Gender_i$ is "1", it indicates that this user is a woman, otherwise this user is a man.

• *SNS involvement*: $NumOri_{it}$, $Interval_{it}$

Like prior studies [60], $NumOri_{it}$ is measured by the total number of tweets authored by the user in the past six months. Retweet interval $Interval_{it}$ is the number of days since the user forwarded a tweet last time. Since the maximum of $NumOri_{it}$ is about 2 thousand, we divide $NumOri_{it}$ by 1.0e3 to make different predictors on the same scale.

**Control variables.** In addition to above-mentioned variables, we also include the following factors as control variables, since they have been identified as influential factors on individuals' sharing behavior in prior research [58, 61, 62].

- *Source attractiveness*:'

Twitter enables users to follow any other accounts at the click of a button (except protected account, meaning that they can be viewed only by approved followers). Thus, the number of an author's followers ($fans_{it}$) signifies how popular and attractive the author is. For the same reason mentioned above, we divide $fans_{it}$ by 1.0e5 to make different predictors on the same scale.

- *Bandwagon effect*: $bandwagon_{it}$

Researchers have confirmed that individuals' retweeting decisions can be affected by other people's reactions. Thus, we employ the number of times that the post has been retweeted to denote bandwagon effect. Since the maximum of $bandwagon_{it}$ is about 6 thousand, we divide $bandwagon_{it}$ by 1.0e3 to make different predictors on the same scale.

Other control variables include the amount of hashtags in a post ($numtags_{it}$), the amount of URLs in a post ($numurl_{it}$) and the length of a post ($length_{it}$).

## Data processing and summary statistics

For the 1,437 core users in our dataset, we have the data every time they retweeted, the huge amount of data points makes STATA intolerably slow to calculate the regression result. Therefore, we downsample each user's behavior once every three days. Finally, we get a panel dataset which consists of 1364 twitterers' retweeting behavior and have 138,862 data points. The cross-section is the user and the time variable is days. Table 1 presents the descriptive statistics of all the variables in the study. Correlation results reveal that the correlation coefficients between each other are weak or moderate.

## Regression model

Considering the dataset is an unbalanced panel and the dependent variable is dichotomous, we use the panel logit model to test the proposed hypotheses, because there isn't fixed effects model for panel probit model. In the current study, we are interested in the effect of topic familiarity on individual retweeting decision and propose the effect is contingent on the moderating effects of social influence and personal control. To test those proposed hypotheses, topic familiarity of a post ($Topfami_{it}$), interaction items between $Topfami_{it}$ and these moderators are added in Eq (2). The moderators are also included as covariates. Besides, according to prior studies [61], individuals' sharing decision is also significantly influenced by source attractiveness ($fans_{it}$), bandwagon effect ($bandwagon_{it}$) and the amount of information in a post ($length_{it}$, $numtags_{it}$, $numurl_{it}$). Thus, we add these control variables into Eq (2).

To relax the assumption of conditional independence among the responses for the same twitterer given the covariates, we add a twitterer-specific random intercept $\zeta_i$ in Eq (2) and obtain a random-intercept logistic regression model. Specifically, when $\zeta_i$ equals $\zeta$ for all the users, the twitterer's individual effect disappears and Eq (2) becomes a pooled logit model. When individual effect exists, there are two situations. The first one is, $\zeta_i$ is uncorrelated with other predictors and Eq (2) becomes a random-effects (RE) panel logit model. The second one is, $\zeta_i$ is correlated with other predictors and Eq (2) becomes a fixed-effects (FE) panel logit model.

**Table 1. Descriptive statistics of all the variables.**

|  | N | Mean | Std. Dev. | min | max | VIF |
|---|---|---|---|---|---|---|
| y | 138862 | .4 | 0.49 | 0 | 1 |  |
| Topfami | 138862 | .14 | 0.11 | 0.02 | .65 | 1.01 |
| Followers[a] | 138862 | .20 | 0.56 | .001 | 3.11 | 1.11 |
| AveTwtLen | 138862 | 12.56 | 3.07 | 5.34 | 19.33 | 1.08 |
| Gender | 138862 | .46 | 0.45 | 0 | 1 | 1.03 |
| NumOri[b] | 138862 | .35 | 0.36 | 0.005 | 1.67 | 1.11 |
| Interval | 138862 | 6.86 | 13.51 | 1 | 86 | 1.04 |
| numurl | 138862 | .75 | 0.67 | 0 | 2 | 1.09 |
| numtags | 138862 | .61 | 1.01 | 0 | 12 | 1.05 |
| length | 138862 | 15.02 | 5.34 | 3 | 27 | 1.07 |
| fans[a] | 138862 | 5.24 | 17.85 | 0.001 | 128 | 1.15 |
| bandwagon[b] | 138862 | .20 | 0.78 | 0.001 | 6.06 | 1.16 |

[a]*Notes*:These variables have been divided by 1.0e5.

[b]*Notes*:These variables have been divided by 1.0e3.

$$
\begin{aligned}
\text{logit}(\text{Pr}(Y_{it} = 1)|\mathbf{x}_{it}, \zeta_i) = {}& \beta_0 + \beta_1 Topfami_{it} + \beta_2 Followers_i \\
& + \beta_{21} Followers_i \cdot Topfami_{it} + \beta_3 AveTwtLen_{it} + \beta_{31} AveTwtLen_{it} \cdot Topfami_{it} + \beta_4 Gender_i \\
& + \beta_{41} Gender_i \cdot Topfami_{it} + \beta_5 NumOri_{it} + \beta_{51} NumOri_{it} \cdot Topfami_{it} + \beta_6 Interval_{it} \\
& + \beta_{61} Interval_{it} \cdot Topfami + \beta_7 numtags_{it} + \beta_8 numurl_{it} + \beta_9 length_{it} + \beta_{10} fans_{it} \\
& + \beta_{11} bandwagon_{it} + \zeta_i
\end{aligned}
\tag{2}
$$

## Results

### Data analysis and results

Table 2 lists regression results of different models. In column (1)~column (3), an FE panel logit model is employed. In the model of column (1), only control variables are included. And we can see that all the control variables have a significant effect on individuals' sharing behavior. And then, we add "topic familiarity" (Topfami) into the model and obtain the result shown in column (2). The coefficient for Topfami ($\beta_1$) is positive and significant ($\beta_1 = 4.67$, p = 0.00). Therefore, H1 is supported; the more familiar a message is perceived, the more likely this individual will share the message. From Table 2, we can see that both AIC and BIC decrease after "topic familiarity" is incorporated into the model. Pseudo $R^2$ increases from 0.022 to 0.058. A Likelihood ratio test shows that Model (2) favors Model (1) with $\chi_2 = -2\ln$ (likelihoodnull model) + 2ln(likelihoodalternative model) = $-2 * (-83340) + 2 * (-80270)$ = 6340. Therefore, we reject the null model, namely Model (1), at the 1‰ level.

Next, all the five moderators and their interaction items are added into the regression. Results are shown in column (3). This time AIC, BIC decrease and Log-likelihood increases remarkably. And the Pseudo $R^2$ increases from 0.058 to 0.103. The result of model (3) shows that regarding online self-presentation efficacy, both "number of followers" ($\beta_{21} = 1.58$, p = 0.00) and "average tweet length"($\beta_{31} = 0.108$, p = 0.00) positively moderate the influence of topic familiarity on individuals' sharing behavior. Therefore, H2a and H2b are supported. Twitterers with a larger number of followers and twitterers who tend to post longer original

**Table 2. Regression results—dependent variable: *Retwt*.**

| | Hypothesis | (1) | (2) | (3) | (4) | (5) |
|---|---|---|---|---|---|---|
| | | FE | FE | FE | RE | Pooled |
| Topfami | H1 | | 4.67*** | 2.561*** | 2.428*** | 1.918*** |
| | | | (.064) | (.296) | (.292) | (.726) |
| Followers | | | | - | .126*** | .075*** |
| | | | | - | (.033) | (.029) |
| Followers*Topfami | H2a | | | 1.58*** | 1.509*** | 1.1*** |
| | | | | (.139) | (.136) | (.187) |
| AveTwtLen | | | | .065*** | .041*** | .02** |
| | | | | (.007) | (.005) | (.009) |
| AveTwtLen*Topfami | H2b | | | .108*** | .114*** | .117** |
| | | | | (.022) | (.022) | (.048) |
| Gender | | | | - | .037 | .069 |
| | | | | - | (.043) | (.054) |
| Gender*Topfami | H3a | | | .153 | .176 | .104 |
| | | | | (.158) | (.155) | (.314) |
| NumOri | | | | -.9*** | -.667*** | -.137** |
| | | | | (.037) | (.035) | (.062) |
| NumOri*Topfami | H3b | | | .961*** | .936*** | .814** |
| | | | | (.171) | (.169) | (.379) |
| Interval | | | | -.069*** | -.068*** | -.062*** |
| | | | | (.002) | (.002) | (.003) |
| Interval*Topfami | H3c | | | -.016* | -.021** | -.03** |
| | | | | (.01) | (.01) | (.015) |
| numurl | | .371*** | .367*** | .388*** | .382*** | .355*** |
| | | (.01) | (.01) | (.01) | (.01) | (.017) |
| numtags | | .159*** | .172*** | .175*** | .173*** | .146*** |
| | | (.006) | (.007) | (.007) | (.007) | (.009) |
| length | | -.006*** | -.013*** | -.013*** | -.014*** | -.014*** |
| | | (.001) | (.001) | (.001) | (.001) | (.002) |
| fans | | .001*** | .002*** | .003*** | .002*** | .002*** |
| | | (.0004) | (.0004) | (.0004) | (.0004) | (.001) |
| bandwagon | | .268*** | .264*** | .269*** | .268*** | .247*** |
| | | (.009) | (.009) | (.009) | (.009) | (.015) |
| Observations | | 138862 | 138862 | 138862 | 138862 | 138862 |
| Pseudo $R^2$ | | .022 | .058 | .103 | - | .102 |
| AIC | | 166690.83 | 160553.35 | 152803.08 | 163388.48 | 167642.56 |
| BIC | | 166740.03 | 160612.4 | 152940.86 | 163565.62 | 167809.86 |
| Log lik | | -83340 | -80270 | -76387 | -81676 | -83804 |

Standard errors are in parentheses.

*** $p < .01$,

** $p < .05$,

* $p < .1$

contents have a higher propensity for maintaining a consistent and focused persona. They achieve this goal by sharing familiar and relevant topics when making retweeting decisions.

We expect that female twitterers are more likely than male twitterers to retweet familiar topics to communicate a more consistent online identity. The result of model (3) shows that

the moderating effect of "Gender" is not statistically significant. That is to say, there are no significant gender differences in their tendency for a more consistent online identity. Therefore, H3a is rejected.

About SNS involvement, we expect that highly involved users are more motivated to build a consistent and clear online identity by retweeting familiar topics. The result of model (3) shows that the coefficient of NumOri *Topfami is positive and statistically significant ($\beta_{51}$ = 0.961, p = 0.00). Another measurement of SNS involvement is retweet interval. From column (3), we see that the moderating effect of "retweet interval" is negative and significant ($\beta_{61}$ = −-0.016, p = 0.084). Therefore, both H3b and H3c are supported. That is to say, active users manifested by a larger number of original tweets and shorter retweet intervals have a higher propensity for a consistent and focused persona. They have a higher level of involvement and identification with the platform and are more motivated to convey a clear and impressive online image via retweeting familiar topics. Detailed information about the regression result of model (3) is listed in Table 3. Table 4 summarizes results of all the hypotheses.

## Robustness check

**The issue of multicollinearity.** We implemented a series of tests to ensure the robustness of our empirical results. First, we calculated variance inflation factors for all the predictors. The results are shown in Table 1. All variance inflation factors were smaller than 1.2, demonstrating no severe multicollinearity concerns in the current research. Therefore, our findings are not significantly impacted by multicollinearity issue.

**Unobserved individual characteristics.** Researchers have revealed that different individuals have different retweeting tendencies. Some users tend to share trending topics with their followers [33], while some users limit the scope of topics in their shared content to present a consistent and clear online image [10, 11]. Thus, individuals' sharing action could be driven by unobserved characteristics. To capture the unobserved heterogeneity among SNS users, we add a user-specific random intercept ($\zeta_i$) in Eq (2).

As a robustness check, we also run a traditional pooled logit model and a RE panel logit model for comparison. Both models include the same variables as the FE model in column (3). The regression results of the RE panel logit model and the pooled logit model are listed in column (4) and column (5) of Table 2, respectively. By comparing the results in these three models, we can see that the polarity and significance of all the coefficients are very consistent. Thus, our findings show consistency on alternative models.

To illustrate the validity of incorporating individual effects ($\zeta_i$) in the model, we carry out a comparative analysis between the traditional pooled logit model and the FE panel logit model. The FE panel logit model in column (3) yields a larger log-likelihood ($\Delta$ = (−76387) − (−83804) = 7417, a 9.7% boost) and lower values of AIC and BIC, suggesting a better fit with the dataset. A Hausman test is used to test the null hypothesis of a pooled logit against the alternative hypothesis of an FE model (H0: $\rho$ = 0), which favors the FE model with $\chi_2$ = 1947.45. Thus, it is necessary to include individual effects ($\zeta_i$) in Eq (2).

To decide whether the FE logit model or the RE logit model is more suitable for this study, we also compare them in a similar way. The log-likelihood of the FE logit model is 6.9% larger ($\Delta$ = (−76387) − (−81676) = 5289, an 6.9% boost) than that of the RE logit model. Moreover, both AIC and BIC are lower in the FE model than those in the RE logit model. A Hausman test is used to discriminate between the FE model and the RE model, which favors the FE model with $\chi_2$ = 919.76. Thus, the FE logit model is preferred over both the pooled logit model and the RE logit model.

**Table 3. Detailed results of model (3) in Table 2.**

| Y | Coef. | St.Err. | t-value | p-value | [95% Conf | Interval] | Sig |
|---|---|---|---|---|---|---|---|
| Topfami | 2.561 | .296 | 8.64 | 0 | 1.981 | 3.142 | *** |
| Followers | 0 | . | . | . | . | . | |
| Followers*Topfami | 1.58 | .139 | 11.37 | 0 | 1.308 | 1.852 | *** |
| AveTwtLen | .065 | .007 | 9.64 | 0 | .052 | .078 | *** |
| AveTwtLen*Topfami | .108 | .022 | 4.91 | 0 | .065 | .152 | *** |
| Gender | 0 | . | . | . | . | . | |
| Gender*Topfami | .153 | .158 | 0.97 | .334 | -.157 | .462 | |
| NumOri | -.9 | .037 | -24.04 | 0 | -.973 | -.827 | *** |
| NumOri*Topfami | .961 | .171 | 5.61 | 0 | .625 | 1.297 | *** |
| Interval | -.069 | .002 | -38.11 | 0 | -.072 | -.065 | *** |
| Interval*Topfami | -.016 | .01 | -1.73 | .084 | -.035 | .002 | * |
| numurl | .388 | .01 | 38.75 | 0 | .368 | .408 | *** |
| numtags | .175 | .007 | 25.64 | 0 | .161 | .188 | *** |
| length | -.013 | .001 | -10.94 | 0 | -.016 | -.011 | *** |
| fans | .003 | 0 | 6.91 | 0 | .002 | .003 | *** |
| bandwagon | .269 | .009 | 29.15 | 0 | .25 | .287 | *** |
| Mean dependent var | 0.398 | SD dependent var | | | | | 0.490 |
| Pseudo r-squared | 0.103 | Number of obs | | | | | 138862 |
| Chi-square | 17601.366 | Prob > chi2 | | | | | 0.000 |
| Akaike crit. (AIC) | 152803.080 | Bayesian crit. (BIC) | | | | | 152940.857 |

*** p < .01,

** p < .05,

* p < .1

However, we need to point out an inherent disadvantages of the FE logit model. The FE logit model cannot estimate the coefficient of time-invariant variables. For example, in column (3), coefficients of both Followers and Gender are zero, as these two variables are constant for the ith individual each time he or she makes the retweeting decisions. Since these two variables are not the focus of our concern, these issues can be ignored in the current study.

**Robustness test on a smaller group of people.** To test the consistency of our results, we analyze a reduced dataset using the FE panel logit model and the RE panel logit model. Any

**Table 4. Summary of results.**

| Hypotheses | | Results |
|---|---|---|
| H1 | The more familiar a tweet is perceived, the more likely an individual will retweet it to all his or her followers on SNS to build a consistent online identity. | Supported |
| H2a | The number of followers positively moderates the influence of topic familiarity on individuals' sharing behavior. | Supported |
| H2b | Average tweet length positively moderates the influence of topic familiarity on individuals' sharing behavior. | Supported |
| H3a | The influence of topic familiarity on individuals' sharing behavior is stronger for female twitterers than for male twitterers. | Not supported |
| H3b | Number of original tweets positively moderates the influence of topic familiarity on individuals' sharing behavior. | Supported |
| H3c | Retweet interval negatively moderates the influence of topic familiarity on individuals' sharing behavior. | Supported |

users who have shared other-sourced tweets less than 100 times are eliminated from the original dataset. The reduced dataset has 604 valid users and consists of 79,484 datapoints. The results are presented in Table 5, where Model (1) and Model (3) used the reduced sample and Model (2) and Model (4) used the original sample. To be succinct, we only present the results of the hypothesized effects in Table 5. By comparing the results on the two samples, we can see that the hypothesized effects are quite consistent on the two datasets. We also notice that the moderating effect of SNS involvement in the reduced sample becomes much larger than that in the original sample. Specifically, the coefficient of NumOri *Topfami and Interval*Topfam in the reduced sample, is almost 3 times larger than those in the original sample in absolute terms.

There is only one exception. The moderating effect of Gender is supported on the reduced sample ($\beta_{41}$ = 0.865, p = 0.00 in FE model; $\beta_{41}$ = 0.703, p = 0.00 in RE model). That is to say, among users who retweeted very actively, male twitterers are more likely than female twitterers to add new topics and convey a knowledgeable online identity. A possible explanation is that these highly active users are willing and able to devote time to cultivate their online identities, among whom men are more willing than women to broaden the span of topics. Accordingly, male users appeal to new friends with different backgrounds, gain access to new social networks, and finally increase their bridging social capital [56]. Compared with these highly active users, casual users who are representative of the Twittersphere population may feel it is difficult and costly to manage multiple identities via retweeting different topics. Therefore, in the original sample, there are no significant gender differences in their tendency for a more consistent online identity.

**Table 5. Robustness checks using a reduced dataset.**

| Variables | | Fixed effects panel logit model (1) | Fixed effects panel logit model (2) | Random effects panel logit model(3) | Random effects panel logit model(4) |
|---|---|---|---|---|---|
| | | Coefficient (reduced sample) | Coefficient (original sample) | Coefficient (reduced sample) | Coefficient (original sample) |
| | | estimate | estimate | estimate | estimate |
| Topfami | H1 | 1.932*** | 2.561*** | 1.913*** | 2.428*** |
| | | (.433) | (.296) | (.425) | (.292) |
| Followers*Topfami | H2a | 1.051*** | 1.58*** | 1.019*** | 1.509*** |
| | | (.187) | (.139) | (.182) | (.136) |
| AveTwtLen*Topfami | H2b | .091*** | .108*** | .092*** | .114*** |
| | | (.033) | (.022) | (.033) | (.022) |
| Gender*Topfami | H3a | .865*** | .153 | .703*** | .176 |
| | | (.213) | (.158) | (.208) | (.155) |
| NumOri*Topfami | H3b | 3.131*** | .961*** | 3.149*** | .936*** |
| | | (.322) | (.171) | (.317) | (.169) |
| Interval*Topfami | H3c | -.077*** | -.016* | -.086*** | -.021** |
| | | (.025) | (.01) | (.025) | (.01) |
| AIC | | 90343 | 152803.08 | 95210 | 163388.48 |
| BIC | | 90473 | 152940.86 | 95377 | 163565.62 |
| Log like | | -45157 | -76387 | -56538 | -81676 |
| Pseudo R$^2$ | | 0.092 | 0.103 | - | - |
| N | | 79484 | 138862 | 79484 | 138862 |

## Discussion

### Key findings

Considering the practical value of users with a more consistent and clear online identity to social media platforms and marketers, we examine individuals' behavior of constructing online identities through retweeting on Twitter from the perspective of online identity and self-presentation. Specifically, we investigate the moderating effects of online self-presentation efficacy and social influence on individuals' desire for a consistent identity. A framework is proposed and validated based on data from 1364 Twitter users. Several important findings are obtained from the current research. First, most SNS users are prone to retweet familiar topics to communicate a consistent and focused online identity. This finding agrees with the conclusion of [11], which claims that Twitter users tend to retweet the topics they are able to self-produce in order to construct a clear online persona. It also echoes the finding of [9]. They prove that compared with narrowcasting which is recipient focused, broadcasting is motivated by self-presentation motives and makes people to share more self-presentational content.

Second, we confirm that both online self-presentation efficacy and social influence moderate the relationship between topic familiarity and individuals' retweeting behavior.. Prior research reveals that expert users, namely professional bloggers who usually have lots of followers on Twitter, are more likely to convey a consistent online identity [11]. Our research extends this study and shows that users with higher online self-presentation efficacy (not limited to experts), manifested by a larger number of followers and longer original tweets, exhibit a stronger than average propensity to maintain a consistent online identity and thus are more likely to retweet familiar topics. These findings confirm the impact of self-efficacy on people's behavior [51] and respond to [63]'s call for research on "how audience size moderates what people share". According to our findings, Twitter users with a larger audience are more likely to share familiar topics to convery a consistent and focused online identity.

As for social influence, the significant moderating effect of SNS involvement confirms and extends the study of [45] which reveals the influence of identification on people's identity-related behavior in the physical world. We validate that users who are more involved with the social networking site, manifested by a larger number of original tweets and a shorter retweet interval, have a stronger than average tendency to build a consistent online identity through sharing familiar topics. This complies with prior findings that highly-involved community members have a positive attitude toward the community and a stronger intention to stay within the community [64]. Therefore, they care more and spend more effort in establishing their online images than average users [23].

In general, the moderating effect of gender is not statistically significant. That is to say, when making retweeting decisions, male twitterers are not more prone than female twitterers to add new topics and communicate a knowledgeable and diverse online identity. We conjecture that it requires time, patience and cognitive resources to gain a deep understanding of a new topic. Therefore, for casual users, it is costly and time-consuming to successfully manage multiple identities through retweeting topics in multiple areas. In support of this, prior researchers analyze how adolescents use weblogs to construct their identities and find that the blogs created by both genders are more alike than different, and both genders choose to establish a stable and cohesive online identity to represent who they are [65, 66]. However, we also find that among users who are highly active in retweeting other-sourced tweets, male twitterers are more likely than female twitterers to add new topics and convey a diverse online identity, in order to appeal to a wider range of new friends [56].

## Theoretical implications

Our study makes several important theoretical implications. First, we extend self-presentation theory by applying it in the context of individuals' retweeting behavior on Twitter, paving the way for future research along the similar line. Furthermore, by integrating self-presentation theory with social influence theory and social cognitive theory, we reveal the modearting effect of online self-presentation efficacy and social influence on individuals' desire for a consistent online identity. Finally, we propose new variables that gauge individuals' online self-presentation efficacy and SNS involvement. We elaborate on these implications in the paragraphs below.

Existing reseach tends to explain individuals' retweeting motivations using the Information Adoption Model (IAM) [67], Elaboration Likelihood Model (ELM) [10], the Heuristic-Systematic Model (HSM) [68], or diffusion of innovations theory [69]. However, it should be pointed out that these theories are originally developed to investigate people's information adoption behavior, which is quite different with the retweeting behavior. Specifically, information adoption is a more passive and inner behavior (e.g., accept the suggestions or being persuaded), while retweeting behavior is a sharing behavior which is more active and outreach involving information transmission. To support the theoretical framework and develop our hypotheses, this research proposes to explain individuals' retweeting behavior from the perspective of online identity and self-presentation. Self-presentation theory has been used to explain individuals' online behaviors such as interacting with peers on SNS [12], purchasing digital items [19], contributing knowledge [7, 22], brand-related word-of-mouth [70] and so on. In the current study, we use self-presentation theory as the overarching theoretical framework to study individuals' retweeting behavior on Twitter. Therefore, we extend the applicability of self-presentation theory and pave the way for future research focusing on people's retweeting behavior on SNS.

Furthermore, to the best of our knowledge, our study is the first to study the moderating variables which affect people's desire for a more consistent and focused online identity. This is important, as extensive research has proven that online identity positively affects users' satisfaction and continuous usage intention of social networking sites [5, 15, 16, 68], and also relates to consumptive behaviors [4, 71] and eWOM behaviors on the platforms [72]. To identify which user groups have a stronger tendency for a consistent online identity, we extend self-presentation theory by integrating it with social cognitive theory and social influence theory. This novel conceptual framework allows us to explore the moderating effects of online self-presentation efficacy and involvement with the social media platform on individuals' tendency for a more consistent and clear online identity.

Finally, we propose to measure an individual's online self-presentation efficacy using "the number of followers" and "average tweet length", and to measure an individual's level of SNS involvement using "the number of original tweets" and "retweet interval". Future research which relies on real dataset from certain social networking sites can consider these measurements.

## Practical implications

Users' sharing behaviors such as publishing original articles or relying information to a larger audience is vital to the prosperity of social networking sites. Many social networking sites cannot motivate their members toward ongoing sharing behaviors and thus fail to realize their true potential. Our research reveals that individuals are inclined to build a consistent online identity, which motivates them to share familiar and relevant messages. Therefore, social networking sites should help users explore their "real selves" such as their tastes, attitudes and preferences.. For active participants who frequently produce original posts or retweet other-

sourced tweets, social networking sites could design and reward virtual badges to confirm their expertise in a certain area and their voluntary contribution to the platform. For passive members who only browse and seldom contribute to the platform, social networking sites could construct a profile for them from time to time, based on the user's traces on the platform, such as the communities they join in, the frequencies they visit the platform, topics they have "liked", etc. These digital elements give timely feedback to users, help users know themselves better and bring them recognition and an amplified sense of self-worth, which may enhance users' loyalty and contribution to the social media platform.

Our research also offers several suggestions for practitioners who want to employ influencer marketing on social media. First, when selecting voluntary endorsers, make sure that there is a high congruence between the endorser and the product or the brand. This is because driven by their online identities, endorsers are prone to share identity-congruent messages and divert attention away from identity-inconsistent information. Existing research also shows that high influencer-product congruence prompts high audience-product congruence, leading to favorable attitudes toward the product and higher purchase and sharing intentions. Second, among the candidates of endorsers, users who are characterized by (1) owning a larger number of followers, (2) authoring longer and more original tweets than average, (3) being active in retweeting other-sourced posts, are more likely than others to share identity-congruent messages, because they have a stronger tendency for a consistent online identity. Companies should give higher priority to these users when selecting influential endorsers. Social media platforms should also manage to retain these valuable users as they have the potential to become KOLs and can attract more followers [19].

## Limitations and future research

This study has several limitations which can be investigated in future research. First, our study is limited by the fact that the data were collected from only one social networking site. In recent years, new kinds of social media platforms have emerged and they differ greatly from one another in several aspects. For example, on Facebook, users can use friend lists to manage their friends. The extent to which close friends, acquaintances and restricted friends get access to your updates are different. This is different with Twitter where all your followers will see what you have published or retweeted. It would be interesting to examine how structural differences affect users' online identities, whether there are differences among users' online identities within different groups and how they relate to each other.

Second, it would be interesting to investigate the moderating effect of sender-receiver relationship on the relationship between topic familiarity and individuals' retweeting behavior. Given the same degree of topic familiarity, are individuals more likely to retweet a tweet from a strong social tie or a weak social tie? What's the underlying mechanism behind the moderation effect? These questions are worth further exploration and will give guidances on the selection of endorsers for companies.

Finally, the current study reveals that individuals are prone to convey a consistent and clear online identity on social networking sites. Several interesting questions might be raised revolving around one's online identity. For example, can you infer how old someone is, or determine their gender and race based on their online identities. Can you speculate their scores on the Big Five personality scale (i.e., neuroticism, extraversion, openness to new experience, agreeableness, and conscientiousness)? The answers to these questions have great implications for both academians and practitioners.

## Supporting information

**S1 Data.**
(DTA)

## Author Contributions

**Conceptualization:** Juan Shi.

**Data curation:** Juan Shi, Gang Chen.

**Formal analysis:** Juan Shi, Kin Keung Lai.

**Funding acquisition:** Juan Shi.

**Investigation:** Juan Shi.

**Methodology:** Juan Shi.

**Software:** Gang Chen.

**Supervision:** Kin Keung Lai.

**Validation:** Juan Shi.

**Writing – original draft:** Juan Shi.

**Writing – review & editing:** Juan Shi.

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
