## [Decision Letter · Decision Letter 0]

7 Dec 2022

PONE-D-22-23963Examining retweeting behavior from the perspective of self-presentation on social networking sitesPLOS ONE

Dear Dr. Shi,

Thank you for submitting your manuscript to PLOS ONE. After careful consideration, we feel that it has merit but does not fully meet PLOS ONE’s publication criteria as it currently stands. Therefore, we invite you to submit a revised version of the manuscript that addresses the points raised during the review process.

 The paper should be revised in order to solve all the issues highlighted by the reviewers. In particular, it is important to update the dataset which is not updated.

We look forward to receiving your revised manuscript.

Kind regards,

Barbara Guidi

Academic Editor

PLOS ONE

Journal Requirements:

2. Please change "female” or "male" to "woman” or "man" as appropriate, when used as a noun (see for instance https://apastyle.apa.org/style-grammar-guidelines/bias-free-language/gender).

3. In your Methods section, please include additional information about your dataset and ensure that you have included a statement specifying whether the collection and analysis method complied with the terms and conditions for the source of the data.

Reviewers' comments:

Reviewer's Responses to Questions

**Comments to the Author**

1. Is the manuscript technically sound, and do the data support the conclusions?

Reviewer #1: Yes

Reviewer #2: Yes

2. Has the statistical analysis been performed appropriately and rigorously? 

Reviewer #1: No

Reviewer #2: Yes

3. Have the authors made all data underlying the findings in their manuscript fully available?

Reviewer #1: Yes

Reviewer #2: Yes

4. Is the manuscript presented in an intelligible fashion and written in standard English?

Reviewer #1: Yes

Reviewer #2: Yes

5. Review Comments to the Author

Reviewer #1: Although the paper is interesting there are several issues that needs to be addressed.

1. The abstract is not informative and needs to be improved.

2. How was the random "we randomly selected 1250 members of Twitter" done?

3. Why the period selected was 2013-2016 which maybe be outdated since with eh onset of the pandemic the constructs of interest may have changed in terms of importance.

4. Why is there a need for "log transformation".

5. There is no need for mean centering in moderation.

6. When the beta value is not significant you should not interpret the sign as it is 0.

7. The reporting rigor needs to be improved as currently there is no std error, t-values, p-values, confidence intervals.

8. Discussion is very weak and does not do a good job of comparing and contrasting with the literature.

9. Implications are very weakly developed.

10. There was no discussion of changes in explanatory power with the inclusion of the interaction effects.

11. The review is also mostly dated as old as the time the data was collected which raises another issue of applicability.

12. Relook at the limitations as some the limitations maybe raise doubt about the validity and reliability of the findings.

Reviewer #2: The paper proposes an analysis concerning the retweeting behavior from the perspective of self-presentation on social networking sites. The topic of the paper is interesting, but the paper has some research issues that should be faced in order to be published.

In particular, the paper should be revised in order to improve the readability. Furthermore, the main issue concerns the dataset. It seems to be limited and not up to date, and this is a big issue considering the behaviour of users that is completely changed during the years. The dataset should be reconsidered, and this is important to decide if the paper could be accepted or not.

6. PLOS authors have the option to publish the peer review history of their article (what does this mean?). If published, this will include your full peer review and any attached files.

Reviewer #1: No

Reviewer #2: No

---

## [Author Response · Author response to Decision Letter 0]

28 Feb 2023

5. Review Comments to the Author

Reviewer #1: Although the paper is interesting there are several issues that needs to be addressed.

1. The abstract is not informative and needs to be improved.

[answer]:

Thank you very much for your careful reading and valuable advice. We have rewritten the abstract. Now it is well organized and much clearer than the old version.

2. How was the random "we randomly selected 1250 members of Twitter" done?

[answer]:

Thank you very much for raising this question. We realize that the previous introduction about data collection process is too simple. During the revision period, we crawled a new dataset from Twitter via Twitter REST API. Now we have rewritten Section 4.1 and introduce the data collection process in detail. Thank you again for your professional question!

3. Why the period selected was 2013-2016 which maybe be outdated since with eh onset of the pandemic the constructs of interest may have changed in terms of importance.

[answer]:

Thank you very much for your opinions on this issue. Now we have updated the data in the paper. We find that although people’s topics of interest may change due to the outbreak of the pandemic, climate change and other issues, what they often forwarded in most cases is still what they are concerned about or what they are familiar with. Detailed data analysis results and robustness check have been updated in the revised manuscript.

4. Why is there a need for "log transformation".

[answer]:

Thank you very much for your careful reading and insightful question! The reason why we log transform some predictors in the old manuscript is, there are great differences among the predictors in terms of their order of magnitude, which are caused by their definitions. For example, “topical familiarity” varies within the range [0,1], however, the maximum of “fans” is more than 80 million. If we do nothing about the scale problem, the coefficients of predictors with very large magnitude will be close to 0. 

A useful method which is always employed to tackle this problem in previous studies is log transformation. By doing this, the distribution of the predictor is close to normal distribution and the differences in their scales become much smaller than before, which can be found from Table 1 in the old manuscript. Therefore, we log transform some predictors (e.g., “Followers”, “bandwagon”) in the old manuscript. However, it is not straightforward to interpret the coefficients after the predictors are log transformed. Besides that, it will also lead to other problems such as multicollinearity.

Therefore, we change our way of dealing with the scale problem in the revised manuscript. We divide predictors with large magnitude by 1.0e5 or 1.0e3. By doing this, all the predictors are on the same scale. Please refer to Table 2 and Table 3 for regression results on the new dataset.

5.There is no need for mean centering in moderation.

[answer]:

Thank you very much for your careful reading and valuable suggestions! We admit that there is no need for mean centering in moderation. Therefore, in the revised manuscript, we did not mean center the independent variable and the moderators.

6. When the beta value is not significant you should not interpret the sign as it is 0.

[answer]:

Thank you very much for your helpful suggestions! Now in the revised manuscript, we did not interpret the sign of any insignificant coefficient any more.

7. The reporting rigor needs to be improved as currently there is no std error, t-values, p-values, confidence intervals.

[answer]:

Thank you very much for your careful reading and helpful suggestions! We have added Table 3 to show detailed information about the fixed effect panel logit model, including std error, t-value, p-value, and confidence intervals. In Table 2, the std err of all the coefficients are in parentheses.

8.Discussion is very weak and does not do a good job of comparing and contrasting with the literature.

[answer]:

Thank you very much for your valuable comments! We admit that the previous discussion does not do a good job. We have rewritten the whole section, including key findings, implications and limitations. Now, we compare our findings and previous research in each paragraph of section 6.1 “Key Findings”. Our findings are highlighted and the logic is much clearer.

9.Implications are very weakly developed.

[answer]:

Thank you very much for your patience and insightful suggestions! We admit that the previous implications are very shallow and not well developed. We have rewritten the 

theoretical implications and practical implications. Now the quality of this part is greatly improved. Thank you again for your careful reading!

10.There was no discussion of changes in explanatory power with the inclusion of the interaction effects.

[answer]:

Thank you very much for your reminding! We have added a new row at the bottom of Table 2, which shows the changes in Pseudo R2 when new variables are added into the model. From Table 2, we can see that, after the moderators and corresponding interaction items are added into the model, the Pseudo R2 increases from 0.058 to 0.103. We also added this finding in “5.1 Data Analysis and Results” (line 438).

11.The review is also mostly dated as old as the time the data was collected which raises another issue of applicability.

[answer]:

Thank you very much for your careful reading and helpful suggestions! Now, we have updated almost all the literature in the paper, especially section 2 “Literature Review”. You many notice that some literature is still remained although they were published many years ago. The reason is these papers are very influential in this area and highly related with our work. For example:

(4)Kim HW, Chan HC, Kankanhalli A. What motivates people to purchase digital items on virtual community websites? The desire for online self-presentation. Information systems research. 2012;23(4):1232-45.

(7)Ma M, Agarwal R. Through a glass darkly: Information technology design, identity verification, and knowledge contribution in online communities. Information systems 

research. 2007;18(1):42-67.

12.Relook at the limitations as some the limitations maybe raise doubt about the validity and reliability of the findings.

[answer]:

Thank you very much for your careful reading and helpful suggestions! Now, we have rewritten this part. Hopefully, no such doubts will arise now.

Reviewer #2: The paper proposes an analysis concerning the retweeting behavior from the perspective of self-presentation on social networking sites. The topic of the paper is interesting, but the paper has some research issues that should be faced in order to be published.

In particular, the paper should be revised in order to improve the readability. Furthermore, the main issue concerns the dataset. It seems to be limited and not up to date, and this is a big issue considering the behaviour of users that is completely changed during the years. The dataset should be reconsidered, and this is important to decide if the paper could be accepted or not.

[answer]:

Thank you very much for your careful reading and helpful suggestions! Now, we have have updated the data in the paper. The validation results of hypotheses are basically consistent with those in old manuscript. Arguably, although people’s topics of interest may change due to the outbreak of the pandemic, climate change and other issues, what they often forwarded in most cases is still what they are concerned about or what they are familiar with.

In addition, we have updated almost all the literature, rewritten the abstract and the discussion. Thank you again for your patience and helpful comments!

---

## [Decision Letter · Decision Letter 1]

10 May 2023

Examining retweeting behavior on social networking sites from the perspective of self-presentation

PONE-D-22-23963R1

Dear Dr. Shi,

We’re pleased to inform you that your manuscript has been judged scientifically suitable for publication and will be formally accepted for publication once it meets all outstanding technical requirements.

Kind regards,

Barbara Guidi

Academic Editor

PLOS ONE

Additional Editor Comments (optional):

Reviewers' comments:

Reviewer's Responses to Questions

**Comments to the Author**

1. If the authors have adequately addressed your comments raised in a previous round of review and you feel that this manuscript is now acceptable for publication, you may indicate that here to bypass the “Comments to the Author” section, enter your conflict of interest statement in the “Confidential to Editor” section, and submit your "Accept" recommendation.

Reviewer #1: All comments have been addressed

Reviewer #2: All comments have been addressed

2. Is the manuscript technically sound, and do the data support the conclusions?

Reviewer #1: Yes

Reviewer #2: Yes

3. Has the statistical analysis been performed appropriately and rigorously? 

Reviewer #1: Yes

Reviewer #2: Yes

4. Have the authors made all data underlying the findings in their manuscript fully available?

Reviewer #1: Yes

Reviewer #2: Yes

5. Is the manuscript presented in an intelligible fashion and written in standard English?

Reviewer #1: Yes

Reviewer #2: Yes

6. Review Comments to the Author

Reviewer #1: (No Response)

Reviewer #2: The dataset has been updated, so there are no other comment to address and the paper can be accepted as it is.

7. PLOS authors have the option to publish the peer review history of their article (what does this mean?). If published, this will include your full peer review and any attached files.

Reviewer #1: No

Reviewer #2: **Yes: **Barbara Guidi

---

## [Editor Report · Acceptance letter]

12 May 2023

PONE-D-22-23963R1 

Examining retweeting behavior on social networking sites from the perspective of self-presentation 

Dear Dr. Shi:

I'm pleased to inform you that your manuscript has been deemed suitable for publication in PLOS ONE. Congratulations! Your manuscript is now with our production department. 

Kind regards, 

on behalf of

Dr. Barbara Guidi 

Academic Editor

PLOS ONE